# Gender Inequalities in Publications about COVID-19 in Spain: Authorship and Sex-Disaggregated Data

**DOI:** 10.3390/ijerph20032025

**Published:** 2023-01-22

**Authors:** Marta Jiménez Carrillo, Unai Martín, Amaia Bacigalupe

**Affiliations:** 1Social Determinants of Health and Demographic Change, Opik Research Group, University of the Basque Country (UPV/EHU), 48080 Leioa, Spain; 2Doctoral Program in Public Health, Department of Preventive Medicine and Public Health, University of the Basque Country (UPV/EHU), 48080 Leioa, Spain; 3Department of Sociology and Social Work, University of the Basque Country (UPV/EHU), 48080 Leioa, Spain

**Keywords:** gender, inequalities, COVID-19, authorship, disaggregation by sex

## Abstract

Gender inequalities in biomedical literature have been widely reported in authorship as well as the scarcity of results that are stratified by sex in the studies. We conducted a bibliometric review of articles on COVID-19 published in the main Spanish medical journals between April 2020 and May 2021. The purpose of this study was to analyse differences in authorship order and composition by sex and their evolution over time, as well as the frequency of sex-disaggregated empirical results and its relationship with the author sex in articles on COVID-19 in the main Spanish biomedical journals. We identified 914 articles and 4921 authors, 57.5% men and 42.5% women. Women accounted for 36.7% of first authors and for 33.7% of last authors. Monthly variation in authorship over the course of the pandemic indicates that women were always less likely to publish as first authors. Only 1.0% of the articles broke down empirical results by sex. Disaggregation of results by sex was significantly more frequent when women were first authors and when women were the majority in the authorship. It is important to make gender inequalities visible in scientific dissemination and to promote gender-sensitive research, which can help to reduce gender bias in clinical studies as well as to design public policies for post-pandemic recovery that are more gender-equitable.

## 1. Introduction

Despite the increase in the participation of women in academia and in scientific research over the last 50 years, gender inequalities among academics and researchers are still evident [1]. Among other things, this is related to the “glass ceiling” effect [2], that is, the invisible and artificial barriers that, along with structural gender inequalities [3], have held back women’s careers. The construct of the “glass ceiling” is linked to another phenomenon, the “leaky pipeline” [4], which reflects the fact that over their professional careers women tend to hold positions with less stability, less prestige, and lower salaries than men due to the presence of institutional barriers, the influence of stereotypes, and personal and other forms of discrimination. A recent review confirmed that the gender gap in medical schools persists worldwide, and that men are almost three times more likely to be full professors [5]. As for the gender composition of the editorial boards of scientific journals, a 2021 study of the world’s 10 most prestigious medical journals found that women held only 21% of the highest positions [6].

This structural gender inequality also affects the intensity of scientific publication [7]. As far as authorship order is concerned, women tend to occupy lower valued positions [8]; first and last authors tend to be men, irrespective of their actual contribution [9]. Moreover, the inclusion of gender perspectives in the different phases of the research process remains very limited [10]. A requirement that is indispensable (though not in itself sufficient) for the incorporation of a gender perspective is the disaggregation of empirical results by sex. A bibliometric review published in Lancet in 2019 [11], based on 11.5 million articles published worldwide between 1980 and 2016, showed that less than 30% of biomedical research studies published their findings disaggregated by sex. That paper also highlighted that a greater presence of women among the authors increased the likelihood of the disaggregation of the results by sex.

COVID-19 has generated a huge body of scientific research. It is important to trace the evolution of gender inequalities in the work published on the pandemic. International evidence suggests that scientific production has fallen among women during this health crisis [12,13,14] probably due to the need to reallocate time, the fact that they tend to provide a higher proportion of care in the family, and the difficulty of achieving an adequate work–family balance [15]. In addition, both the international monitoring of the pandemic [16] and the majority of scientific research into COVID-19 have presented a low level of disaggregation by sex [17], despite the growing evidence of important differences about sex and gender both in SARS-CoV2 infection and its complications [18] and in the effects of vaccines against COVID-19 [19].

Since the beginning of the pandemic, Spain has been one of the countries of the European Union (EU) with the highest mortality from COVID-19. The country has a Gender Equality Index of 73.7—that is above the EU average of 68.0 [20]. However, in academia, in 2019 only 22.5% of full university professors were women [21], and in professional bodies and scientific societies the positions of greater executive power (for example, presidency and deanship) were for the most part held by men [22]. In the field of science, according to a recent report from the Ministry of Science, men account for 59% and women for 41% [23] of all scientists, while in medicine, women already outnumber men (52.5% and 47.15% respectively) [24]. Finally, in the political sphere, during the pandemic, women were clearly underrepresented in expert management committees in Spain [25].

We analysed the differences in authorship order and composition by sex of articles on COVID-19 published in the main biomedical journals in Spain between April 2020 and May 2021. We also assessed the frequency of sex disaggregation of the data analysed in the articles, and its relationship with the authors’ sex composition and order.

## 2. Materials and Methods

Cross-sectional study of the 24 main Spanish medical journals indexed on the SCImago Journal & Country Rank portal [26]. The bibliometric review was based on the PubMed database between April 2020 and May 2021. The MeSH terms “SARS-CoV-2” OR “COVID” OR “COVID-19” OR “coronavirus” were used as descriptors.

A total of 914 publications and 4921 authors were identified. Since one of the journals did not provide any results in the bibliometric search, 23 journals were included in the final analysis.

The manual coding of authors by sex was carried out based on names, using as support the Spanish National Institute of Statistics’ database of the names of the population by sex. In the case of non-Spanish authors, we used the gender detention tool “Genderize.io”. In 23 cases (0.5%), the sex of the author could not be identified. We calculated various indicators to assess the total differences according to sex and according to the author position (i.e., whether an author signed first or last), and the female/male (F/M) ratio was calculated for each position. Next, we examined trends in authorship and in authorship position (again, whether authors signed either first or last vs. middle) in the papers published each month. The month was determined based on the date of publication online.

For our analysis of the frequency of disaggregation by sex in the presentation of empirical results, we excluded letters to the editor, editorials, articles on specific gender-related themes or dealing with a specific sex (e.g., urological, gynecological or obstetrical papers), clinical protocols and consensuses (*n* = 195), and clinical case reports (*n* = 162). Among clinical case articles we calculated percentages of cases based on males, females or both (males and females). Next, we categorised the remaining articles (*n* = 557) in terms of the disaggregation by sex of data, using the following classification: (0) articles that did not consider the sex variable in any case; (1) articles that included the sex variable only in the description of the composition of the sample; (2) articles that used sex as an explanatory or confounding variable in their results; (3) articles that included sex as a stratification variable for the results. Finally, we examined the relationship between the sex of the authors and the sex disaggregation of empirical data. The dependent variable was the sex disaggregation of data irrespective of the degree of disaggregation (categories 1 + 2 + 3 were grouped as “sex variable included in the analysis” and compared with the articles with no disaggregation at all by sex (category 0)). We constructed several independent variables related to the author sex composition: sex of the first author, sex of the last author, and proportions of male and female (mostly male—70% or more of authors, balanced—more than 30% of each sex, or mostly female—70% or more of authors). We also created a joint variable of the combination of the sex of the first and last author. The joint variable included four different categories according to the sex of the first and last author: 1. male–male, 2. male–female, 3. female–male, and 4.female–female. We summarised the association between these variables and whether or not the results were disaggregated by sex by odd ratios based on logistic regression models that were adjusted for the number of authors and the discipline of the journal. Only articles with more than one author were considered in this analysis.

## 3. Results

A total of 914 articles were identified, corresponding to 4921 total authors. Table 1 shows the inequalities between men and women in various indicators regarding the order and composition of the authors of these papers. Women accounted for 42.2% of authors, representing a female/male ratio of 0.74, but for only 36.7% of first authors, and for only 33.7% for last authors. Both first and last authors were men in 43.5% of the articles, but were women in only 15.1% of cases. All authors were male in 21.4% of the articles, and all female in 8.1%. In articles written by a single author, 80.3% were men and 19.7% were women.

The analysis by journal (Table 2) shows that most (65.2%) had an F/M ratio below one. Only 30.4% of the total number of journals had a greater proportion of female authors (F/M ratio > 1).

The trend analysis conducted over the course of the pandemic showed a higher representation of male authors both as a whole and for first and last authors (Figure 1). Among first authors, the lowest F/M ratios were recorded between May and November 2020 (between 0.41 and 0.53), and later decreased. Among last authors, we observed higher volatility and no clear trend over the months, but minimum F/M ratios of up to 0.33.

As regards the disaggregation of the results according to sex (Figure 2), 62% did not consider the sex variable at all and 28% included it only in the description of the sample. Nine percent included sex as an explanatory or confounding variable in the results, and only one percent included it as a stratification variable.

Table 3 shows details of the relationship between the sex of the authors and the sex disaggregation of the data. Both the overall presence of female authors and their participation as first and last authors were associated with a higher rate of sex disaggregation of data in the articles. However, this association was only significant when women were first authors (Model 1) OR = 1.47 [1.03–2.11]). When the last author was a woman (Model 2), the sex variable included in the analysis was not significantly more frequent (OR = 1.29 [0.88–1.88]). The fact that most of the authors were women (Model 3) also favoured disaggregation (OR = 1.64 [0.94–2.68]). When adjusting for the set of variables (Model 4), the association of women as first author and higher rate of disaggregation by sex was proven again (OR= 1.39 [0.95–2.02]).

If we consider both the sex of the first and last author for assessing the relationship between the authors’ sex and sex disaggregation of data (Figure 3), the importance of the first author’s sex is again clear. The association is only significant when the first author is a woman and the last author is a man OR = 1.68 (1.06–2.65), although when the last author is also a woman (*n* = 80) its effect is smaller and not statistically significant OR= 1.64 (0.98–2.77).

Among the clinical cases excluded from the disaggregation analysis (*n* = 162), 47.5% focused on men and 38.8% on women. The remaining 22 clinical cases included more than one subject, with a total of 32 men and 19 women.

## 4. Discussion

The results of this study highlight the notable gender inequalities in the authorship of the articles published in the main Spanish biomedical journals on COVID-19. The differences are even more marked in the case of the main authors (i.e., the first and last authors). Men are more likely to appear as first, last or sole authors. Notable findings are the low degree of sex disaggregation in results, and the greater probability that the sex variable will be considered when the first author of the article is a woman.

The underrepresentation of women as authors in biomedical journals was already reported before the global pandemic [11], and the trend has increased since the outbreak of COVID-19 [27]. Our study is the first in Spain to record this gender gap in authorship during the pandemic. A similar study was performed in 2017 with the same Spanish biomedical journal [28]; as we have not considered publications not related to COVID-19 conducted during the pandemic a total comparison with it is not possible. Despite this universe restriction, the F/M ratio fell both overall (F/M ratio 0.88 in 2017 study) and in 61.0% of journals; it was below one in 65.2%. 

As regards the position of authors according to sex, our results are in line with international evidence indicating that gender inequality is more marked in the authorship positions with the greatest professional impact [29,30]. In the Spanish context, the differences observed in this study regarding the main authorships (first position 36.7% women and last position 33.7% women), are disproportionate to the sex composition of the academic fields targeted in our study, as in the medical field women already outnumber men [24]. Other studies have shown that these differences in the gender distribution of principal authorships also vary depending on geographic area [31]. All of this highlights the importance of applying an intersectional perspective that considers the relationship between different dimensions of social inequality [32].

During the pandemic, the productivity of women academics was lower than that of men [33,34], especially among those who are mothers [15,35], and during the lockdown enforced in response to the first wave [36]. Our results show a gender gap in authorship, especially in the months post-confinement in Spain, and this was more notable with regard to first authors. This fact may be due to the intensification of gender inequalities due to the time traditionally devoted by women to domestic and care work [37,38].

As for the disaggregation of the results of studies of COVID-19 according to sex, our analysis confirms that it remains uncommon [17]. In the Spanish epidemiological monitoring of the pandemic, indicators that would allow an analysis of the intermediate and structural social determinants of inequalities between men and women are few, and so applying a gender perspective to population studies on COVID-19 is limited [39].

Finally, we corroborate previous reports that the sex of the first authors influences the sex disaggregation of results [11]. In contrast, this relationship was not found when analysing the influence of women as last authors; this may be due to their age, since they may belong to generations in which there was little awareness of the gender perspective in research. When jointly considering the sex of the first and last authors, the importance of women as first authors, (but not as last authors), is confirmed again. These results are probably related, not just with the degree of awareness of disaggregating data by sex among the last authors, but also with a smaller sample size, as few articles with women as first and last authors were identified. The fact that there are fewer articles signed by women as both first and last authors is likely related to an under-representation of women in leadership positions in COVID-19 research, which remains more often occupied by men. These findings are in line with another study which also analysed articles on COVID-19 topics specifically [12]. 

Despite the small sample size of the clinical cases analysed (*n* = 162), there seems to be a clear predominance of male-based cases. This has also been observed at the international level [40]. It is plausibly related both to higher cases of COVID-19 complications and mortality among men globally [18], and to a traditionally androcentric construction of science [41].

The main limitation of our study is that the follow-up came to an end in the middle of 2021. Future studies should analyse whether these gender inequalities in authorship were maintained in the following months and years.

The COVID-19 pandemic has confirmed the great challenges posed by the management of public health crises, defined by Sharfstein [42] as follows: recognising or identifying crises and their epidemiological profile, appropriate political management, and public communication. This study has analysed the scientific communication of the pandemic from a gender perspective. This focus is particularly important in order to avoid the reproduction of inequalities between men and women. Scientific journals are a key communication channel for the advancement of knowledge, and so their role in promoting gender equality is paramount [43]. It is essential to continue promoting research incorporating a gender perspective to help break down the historical power dynamics in scientific authorship. During the pandemic, the gender gap in the publication of research by women scientists reflects the persistence of gender inequalities [44] which implies a negative impact on clinical studies.

## 5. Conclusions

With the underrepresentation of women in the scientific, media and political spheres, and the lack of a distinctive discourse and framework of understanding that acknowledges the importance of sex in the evolution of the pandemic, both limit the possibility of generating scientific evidence of the unequal impact of the pandemic on the health of men and women. This study draws attention to the remarkable gender inequalities in the authorship of the articles published in the main Spanish biomedical journals on COVID-19, especially significant in the case of lead authors (i.e., first and last authors) that are more likely represented by men. These results are in line with the same trend that has occurred internationally, especially striking during the lockdown. Furthermore, this research shows the low degree of disaggregation of the results according to sex despite the consequences that these gender biases in COVID-19 research may have on clinical practice. Last but not least, this study have assessed a higher probability that the results will be disaggregated by sex when the first author of the article is a woman. It is important to make gender inequalities visible in scientific dissemination and to promote gender-sensitive research, which would help to reduce gender bias in the clinical approach as well as to design public policies for post-pandemic recovery that are more gender-equitable.

## Figures and Tables

**Figure 1 ijerph-20-02025-f001:**
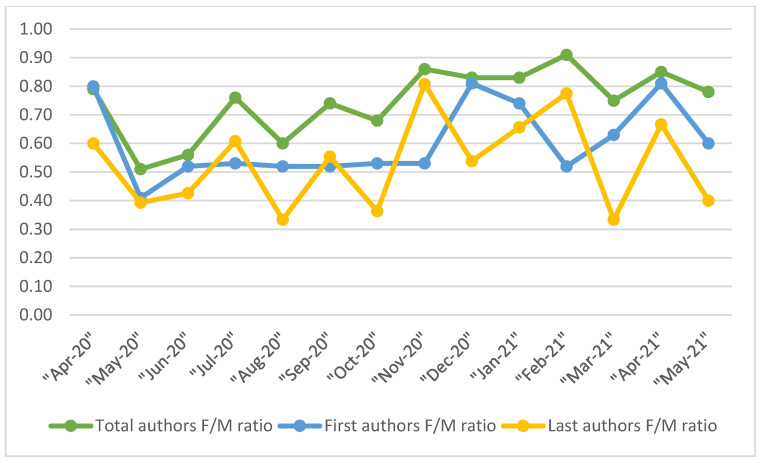
Evolution over time of the female/male (F/M) author ratio for total authors, first authors and last authors, April 2020–May 2021.

**Figure 2 ijerph-20-02025-f002:**
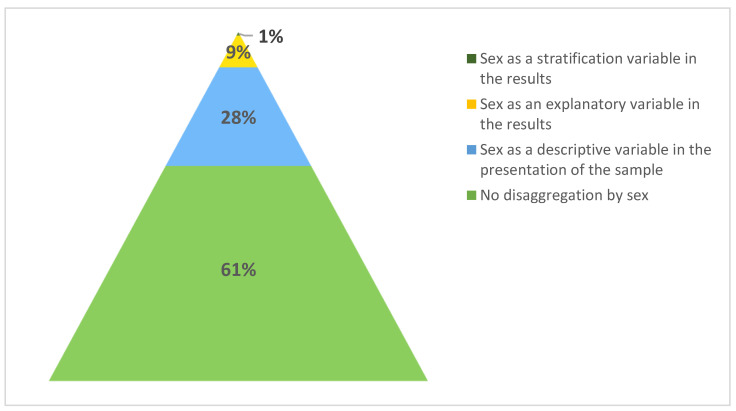
Level of disaggregation by sex of the results in the articles published between April 2020 and May 2021.

**Figure 3 ijerph-20-02025-f003:**
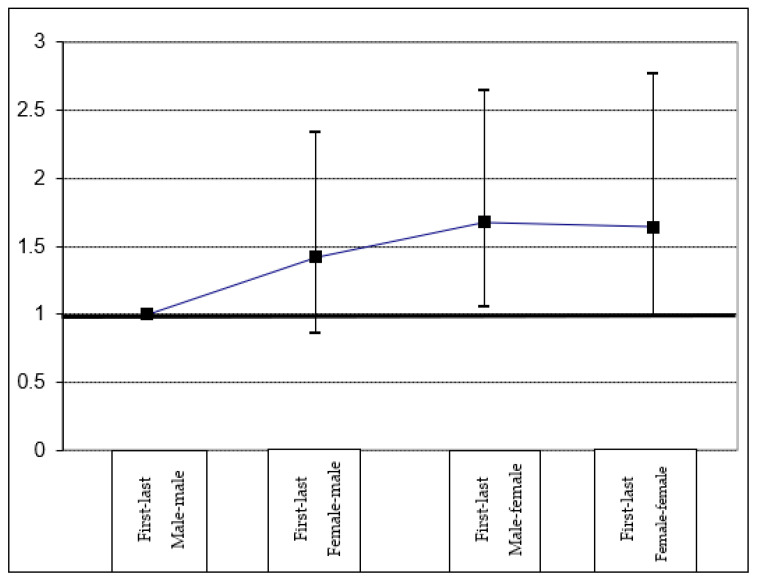
Odd ratios (95% confidence interval) of sex-disaggregation of results according to the combination of the sex of first and last authors.

**Table 1 ijerph-20-02025-t001:** Indicators of author order and composition according to sex (% and female/male (F/M) ratio) of journals between April 2020 and May 2021.

	% Men (*n*)	% Women (*n*)	F/M Ratio
Total authors	57.5 (2815)	42.5 (2083)	0.74
Sex of first author	63.3 (577)	36.7 (335)	0.58
Sex of last author	66.3 (564)	33.7 (287)	0.51
First and last author same sex	43.5 (369)	15.1 (128)	0.35
All authors same sex	21.4 (196)	8.1 (74)	0.38
Articles with single author	80.3 (49)	19.7 (12)	0.25

**Table 2 ijerph-20-02025-t002:** Frequency (% and *n*) of men and women and female/male (F/M) ratio of article authorship according to journal between April 2020 and May 2021.

	% Male Authors (*n*)	% Female Authors (*n*)	F/M Ratio
Acta Otorrinolaringológica Española	66.7 (34)	33.3 (17)	0.50
Actas Españolas de Psiquiatría	13.0 (3)	86.1 (20)	6.67
Anales de Pediatría	45.1 (124)	54.9 (151)	1.22
Archivos de Bronconeumología	59.8 (447)	40.0 (299)	0.67
Archivos Españoles de Urología	79.3 (142)	20.7 (37)	0.26
Atención Primaria	55.0 (99)	44.4 (80)	0.81
Cirugía Española	61.4 (186)	38.3 (116)	0.62
Clínica e Investigación en Arteriosclerosis	69.2 (18)	30.8 (8)	0.44
Endocrinología. Diabetes y Nutrición	43.2 (19)	56.8 (25)	1.32
Enfermedades Infecciosas y Microbiología Clínica	41.0 (16)	59.0 (23)	1.44
Farmacia Hospitalaria	31.1 (28)	67.8 (61)	2.18
Gaceta Sanitaria	53.3 (73)	46.7 (64)	0.88
Gastroenterología y Hepatología	57.3 (82)	41.3 (59)	0.72
Medicina Clínica	56.0 (415)	42.9 (318)	0.78
Nefrología	50.2 (130)	49.8 (129)	0.99
Neurocirugía	20.0 (1)	80.0 (4)	4.00
Nutrición Hospitalaria	58.0 (47)	40.7 (33)	0.70
Reumatología Clínica	58.4 (73)	39.2 (49)	0.67
Revista Española de Cardiología	74.9 (355)	24.9 (118)	0.33
Revista Española de Enfermedades Digestivas	59.3 (118)	39.7 (79)	0.67
Revista Española de Geriatría y Gerontología	49.3 (73)	50.7 (75)	1.03
Revista Española de Quimioterapia	61.1 (182)	38.9 (116)	0.64
Revista Española de Salud Pública	42.5 (150)	57.2 (202)	1.35
Total	57.5 (2815)	42.5 (2083)	0.74

**Table 3 ijerph-20-02025-t003:** Odds ratio (95% confidence interval) of the sex disaggregation of results by variables related to the sex of the authors. Reference categories are indicated with values of 1.

		Model 1	Model 2	Model 3	Model 4
Sex of first author	Male	1			1
	Female	1.47 (1.03–2.11)			1.39 (0.95–2.02)
Sex of last author	Male		1		1
	Female		1.29 (0.88–1.88)		1.09 (0.70–1.67)
Authorship composition	Majority of males			1	1
	Balanced			1.07 (0.63–1.83)	1.13 (0.64–1.98)
	Majority of females			1.64 (0.94–2.89)	1.49 (0.83- 2.68)

## Data Availability

Not applicable.

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
