# Peer review of "Gender Inequalities in Publications about COVID-19 in Spain: Authorship and Sex-Disaggregated Data"

_ijerph, 2023, doi:10.3390/ijerph20032025_

Round 1

Reviewer 1 Report (Previous Reviewer 1)

Based on the authors' response and narrow spectrum of their article (taking into account only Spanish population, which authors justify very vaguely in their response to the review report), I recommend that authors resubmit their article in the Spanish national journal. Therefore, I stand as in my initial review report.

Author Response

Reviewer 2 Report (Previous Reviewer 2)

The authors have answered my comments satisfactorily, and the revised manuscript can be published in IJERPH.

Author Response

Thank you for your valuable and thoughtful comments to our first version of the manuscript and for considering that the revised manuscript can be published in IJERPH.

Round 2

Reviewer 1 Report (Previous Reviewer 1)

I appreciate the authors efforts to revise their manuscript and I also admire their persistance in trying to prove the merits of their work. In authors' reply to the Review Report, they omit one very important argument that was the basis on my decision to suggest, that their manuscript should not be published. In my initial report, I pointed out a very significant flaw, which influences authors results and conclusions. I still stand behind my initial report review, stating that "Differences in the number of scientists of a given gender may also occur due to various reasons: gender discrimination, but also stereotypes, cultural roles, etc. In other words, the lower number of people of a given gender in a given field does not necessarily result from discrimination – sometimes people of a given gender don't want to work in that field. Without taking all those information into account, it is impossible to draw any conclusions from the authors’ analyses.". Authors do not refer to that comment at all. Therefore, I suggest as in in my initial review report, that the presented manuscript should not be published.

Author Response

This manuscript is a resubmission of an earlier submission. The following is a list of the peer review reports and author responses from that submission.

Round 1

Reviewer 1 Report

I want to thank the authors and the Editorial Board for the opportunity to review the article submitted to the International Journal of Environmental Research and Public Health. The authors’ manuscript refers to a very important topic: the gender inequality in COVID-19-related publications in Spain. Unfortunately, I do not believe it is worth publishing in a such reputable journal as IJERPH.

The authors manuscript has some serious limitations. First of all, it is concerned only with Spanish medical journals and it doesn’t take into account non-medical journals and publications of Spanish authors in journals outside Spain.

Second, the statistical analysis is elementary and is focused mainly on percentages and frequencies. Currently, the golden standard of bibliometric analysis is the mapping approach, which could be performed with software such as VOSviewer. The shown results should be analysed more deeply.

Lastly, the article doesn’t take into account one very important variable – how many Spanish scientists are women and how many are men. Without that information, it is impossible to evaluate any gender indifferences in COVID-19 publications because it is unknown if those differences occur due to gender inequality and/or the number of people of a given gender working in science. Differences in the number of scientists of a given gender may also occur due to various reasons: gender discrimination, but also stereotypes, cultural roles, etc. In other words, the lower number of people of a given gender in a given field does not necessarily result from discrimination – sometimes people of a given gender don't want to work in that field. Without taking all those information into account, it is impossible to draw any conclusions from the authors’ analyses.

The authors’ manuscript presents very basic results on a narrow sample, which doesn’t provide anything new to the field outside of Spain. Therefore, due to analytical limitations and the narrow spectrum of the article's field, I do not recommend its publication in IJERPH.

Reviewer 2 Report

This would be a fine study, and it would be publishable after minor changes, if it would not suffer from a fundamental methodological flaw. The study compares the M/F ratio of authors of studies on COVID-19 between April 2020 and May 2021 with the M/F ratio of authors of *all* studies in Spanish medical journals in 2017. It turns out the first is lower, but this begs the question: is it lower because of the period (during the COVID pandemic), or because of the subject (COVID)? The authors favour the first interpretation (and do not mention the second one), and I agree that is quite plausible. But to establish that period effects are behind the result, they would have had to analyze *all* articles in biomedical journals during the COVID period. No reasons are given why only articles with COVID-related MeSH terms were selected. No figure on the proportion of these studies relative to all biomedical studies during this period is given. To remedy this flaw would mean to re-do the research.

Minor issues:

- Manual coding of authors by sex was carried out using as support the Spanish N.I.of S. database of names. But what about non-Spanish names? Or were there no non-Spanish authors?

- It is found that in only 1% of studies the results are disaggregated by sex. This implies that even among female authors, a large majority do not do this. Why? Perhaps they do not think disaggregation by sex of the results is  relevant, or is there another reason?

- Figure 1 on the Evolution over time is not very informative, and could be removed (saving much space). A brief description in the text would be sufficient.